# The Management of Poststroke Thalamic Pain: Update in Clinical Practice

**DOI:** 10.3390/diagnostics12061439

**Published:** 2022-06-10

**Authors:** Songjin Ri

**Affiliations:** 1Department for Neurology, Meoclinic, Berlin, Friedrichstraße 71, 10117 Berlin, Germany; song-jin.ri@charite.de; 2Department of Neurology, Charité University Hospital (CBS), 12203 Berlin, Germany; 3Outpatient Clinic for Neurology, Manfred-von-Richthofen-Straße 15, 12101 Berlin, Germany

**Keywords:** thalamic pain, central pain, stroke, pain, poststroke thalamic pain, central poststroke pain, thalamus, botulinum, deep brain stimulation, opioid

## Abstract

Poststroke thalamic pain (PS-TP), a type of central poststroke pain, has been challenged to improve the rehabilitation outcomes and quality of life after a stroke. It has been shown in 2.7–25% of stroke survivors; however, the treatment of PS-TP remains difficult, and in majority of them it often failed to manage the pain and hypersensitivity effectively, despite the different pharmacotherapies as well as invasive interventions. Central imbalance, central disinhibition, central sensitization, other thalamic adaptative changes, and local inflammatory responses have been considered as its possible pathogenesis. Allodynia and hyperalgesia, as well as the chronic sensitization of pain, are mainly targeted in the management of PS-TP. Commonly recommended first- and second-lines of pharmacological therapies, including traditional medications, e.g., antidepressants, anticonvulsants, opioid analgesics, and lamotrigine, were more effective than others. Nonpharmacological interventions, such as transcranial magnetic or direct current brain stimulations, vestibular caloric stimulation, epidural motor cortex stimulation, and deep brain stimulation, were effective in some cases/small-sized studies and can be recommended in the management of therapy-resistant PS-TP. Interestingly, the stimulation to other areas, e.g., the motor cortex, periventricular/periaqueductal gray matter, and thalamus/internal capsule, showed more effect than the stimulation to the thalamus alone. Further studies on brain or spinal stimulation are required for evidence.

## 1. Poststroke Thalamic Pain

Poststroke thalamic pain (PS-TP), or broadly central poststroke pain (CPSP), is centralized, neuropathic, often characterized with hyperalgesia and allodynia, and is a complaint in 2.7 to 25% of stroke survivors [1,2,3,4,5]. The prevalence of PS-TP was highly variable because of the different study durations as well as the clinical characteristics of the stroke population included in the clinical studies, e.g., stroke etiology, distribution of stroke lesions, and clinical management [1,2,3,4]. PS-TP may develop immediately after a stroke; however, apparent PS-TP often comes later in the postacute phase or the recovery phase following the stroke, e.g., months to years [1,3,4,6], and thalamic stroke may be recalled again later because of the development of PS-TP, which has also been called the Dejerine–Roussy syndrome [1,6,7]. 

Thalamic pain syndrome commonly follows an ischemic stroke or hemorrhagic stroke in the thalamic and lateral medullary areas [3,8,9]. PS-TP occurs after thalamic stroke with sensory deficits, and it often comes up to 40% in the acute phase following the stroke, e.g., within a month of the stroke [10,11,12]. Over 40% of patients with PS-TP have symptoms between one and 12 months after the stroke, and sometimes PS-TP develops between one and six years poststroke [10,12].

Up to 74% of stroke survivors with PS-TP showed gradually increased pain rather than rapid worsening following the stroke, in no relation with the general demographic data and the existence of sensory deficits. Younger age was shown as a risk factor of PS-TP [13].

Right-sided stroke lesions were more commonly associated with the development of PS-TP than left-sided ones. The affected right hemisphere of the brain is better reactive for pain medication [6].

### 1.1. The Pathogenesis of PS-TP

The underlying mechanisms of PS-TP are poorly understood, contributing to challenges in its management. The thalamus, as a relay station for all sensory tracts in the brain, works to decode sensory information and process it, which goes to the somatosensory cortex where it is interpreted [14,15,16] (see Figure 1). Actually, several parts of thalamus, especially the ventrocaudal regions of the thalamus such as the ventral posterior nuclei or the ventral lateral nuclei, the lateral posterior nuclei, as well as the thalamic sensory tracts, e.g., the simultaneously affected spinothalamic tract and the anterior pulvinar nucleus, have been known as the high-risk areas of the development of PS-TP [14,17]. 

The damage to the thalamus or thalamic sensory tracts due to stroke can cause thalamic pain syndrome [14,17,18]. The damage-related changes of the processing and interpretation of peripheral sensory information, e.g., tactile, temperature, pressure, including the loss of them, as well as their malfunction in the afferent pathway from the thalamus to the cortex in central poststroke pain, e.g., tactile or temperature stimuli to the thalamus, can be interpreted as painful (allodynia) or amplifying of painful stimuli, which make it worse (hyperalgesia) [18,19,20,21]. On the other hand, the central sensitization of pain follows due to persistent overactivity [18,20,21].

The pathogenesis of PS-TP has been considered and discussed by several possible theories, such as central imbalance, central disinhibition, central sensitization, other thalamic changes, and inflammatory responses on the affected neural pathway due to stroke [14,17,18,22].

If the spinothalamic tract is damaged due to stroke, and the dorsal–medial lemniscus pathway functions normally, the central imbalance between the abnormal nociception and thermal sensation can occur. This *central imbalance* can also come from third neurons of the spinothalamic pathway, the anatomical projection from the thalamus to the insular cortex, or the anterior cingulate region [14,18].

Due to the stroke affecting the lateral thalamus, this causes *central disinhibition* by the deafferentation of the GABAergic neurons of the ventral posterolateral nucleus, which causes the intrinsic inhibition of the ventral posterolateral nucleus, and the overactivation of cortical areas, developing pain. The delayed appearance of PS-TP may be explained by this. The disinhibition of temperature-sensing fibers (primarily those that sense cold) might be the cause of cold allodynia. Hyperactivities of central afferent neurons, called “*central sensitization*”, can lead pain spontaneously or already on suboptimal stimuli [14,18].

The hypersensitivity of the remaining nerves in the spinothalamic tract, as well as the microglial activation after the stroke, can be associated with the development of PS-TP [18,23]. 

The activation of NLRP3 inflammasome (NOD-like receptor pyrin domain 3; NOD: nucleotide-binding and oligomerization domain) due to inflammatory responses following stroke can cause the decrease of descending fibers from the cortex to the thalamus, which may then induce the reduction of GABAergic release, resulting in the increased excitability of the ventral basal neurons in the thalamus [18]. On the other side, NLRP3 inflammasome at the thalamus lesion strengthens the inflammatory response of microglia at the same time. Persisting inflammatory processes can induce GABAergic changes in the thalamus reticular neurons to inhibit the functions of ventral basal interneurons. These can develope PS-TP [18]. Several factors, e.g., stress and skin temperature, also aggravate PS-TP [11].

### 1.2. The Diagnosis of PS-TP

The first diagnostic step is to suspect the history of thalamic stroke and centralized chronic neuropathic pain [6,21]. Any stroke lesion of the thalamus or the spinothalamic tract should be checked firstly, which has a high risk for the development of PS-TP [14,17].

The diagnosis of poststroke thalamic pain (PS-TP) begins with and is based on the anamnesis, e.g., sensory disorder, tactile and thermal hypersensitivities, and later pain on the entire contralateral half after a thalamic stroke [1,2,24], and on clinical examination, e.g., cold/warm-interpreting errors, severe, constant, or intermittent pain made worse by touching, cold-stimulating, or palpating on the affected side [21]. 

The ventrocaudal regions of the thalamus have been known as the high-risk areas of PS-TP development and include the ventral posterior nuclei or the ventral lateral nuclei, and the lateral posterior nuclei [14,17]. In one cohort study of 42 patients with thalamic stroke, the logistic regression analysis showed that the simultaneously affected spinothalamic tract and the anterior pulvinar nucleus had the highest risk of developing PS-TP [17]. The lesions of the thalamus or the thalamic sensory tracts affected by stroke, which was approved by brain imaging such as MRI or CT, is the most important checkpoint in the diagnosis of PS-TP [14,17,18].

The clinical findings and their timely changes following stroke can be also helpful for its diagnosis. The PS-TP can be spontaneous or evoked in response to external stimuli [21]. The majority of patients complain of constant pain, and only 15% had pain only once daily [21]. Allodynia, e.g., burning pain by cold stimuli, is the most common, in over 60% with PS-TP, and is considered pathognomonic for it [11,25]. Hyperalgesia is also reported frequently in PS-TP [11,26]. Other pains include itching and searing-like [27]. This pain syndrome often correlates with emotional changes, fatigue, cognitive impairment, sleep disorders, and algophobia [28].

The thermal as well as pinprick sensation impairment due to the spinothalamic tract dysfunction is a predictor of the development of PS-TP or CPSP. A full neurological finding, especially proprioceptive sensation, cranial nerves, balance, and speech should also be assessed. The affected area can be colder than others. Temperature and pinprick sensation are partially or completely impaired in patients with PS-TP, while their proprioception and vibration sensation can be intact. [11] Facial and head pain within six months following thalamic stroke, despite no other cause of pain, can mostly predict the development of PS-TP later, and the larger the thalamic lesion, the higher the prevalence of PS-TP, as well as the poorer its prognosis [29].

Neurophysiological tests such as electroencephalogram (EEG) and somatosensory-evoked potential (SEP) examination can be individually performed in the diagnosis of PS-TP [23].

In clinical practice, the diagnosis of PS-TP often requires to be differentiated with chronic pain syndrome, complex regional pain syndrome, syringomyelia, centralized pain syndrome, lateral medullary infarction, multiple sclerosis, and idiopathic peripheral neuropathy [11,19].

Unfortunately, the prognosis of PS-TP is mostly poor. PS-TP are mostly persistent and unchanged. Because of its limited treatment options and the low-evidenced efficacy of different interventions, the management of PS-TP should be individualized and flexible. Early identification and treatment of PS-TP showed more favorable outcomes in a small-sized study, but there is no more evidence for it [13].

## 2. Pharmacologic Treatments of PS-TP

Because of lacking direct causal therapies and the limitation of symptomatic therapies, its management remains still one of the large difficulties in stroke rehabilitation. The PS-TP has been known for many years, but the evidence of its symptomatic management is mostly low or very low, with the difficulties and the outcomes of these treatments often unsatisfactory [30,31]. 

The main therapeutic approaches are to improve tactile and thermal hypersensitivity as well as neuropathic pain and quality of life in patients with PS-TP by using traditional pharmacological treatments and nonpharmacological interventions available for patients with PS-TP. Therefore, the evaluation and treatment of PS-TP often require an interdisciplinary team including a neurologist and pain medicine specialist, and occasionally a neurosurgeon [20,30,31]. 

One recent systematic review assessed the effectiveness and safety of different pharmacological treatments for PS-TP, but their outcomes showed no beneficial effects, with very low to low evidence of all included studies [30]. However, traditional medications as the first and second options in the management of PS-TP helped to improve symptoms in some case reports/several studies [11,30,31,32] Although the recent systematic review and meta-analysis showed limited evidence for their effectiveness, typically applied medications such as antidepressants (e.g., amitriptyline, trazodone, and venlafaxine), anticonvulsants (e.g., gabapentin, pregabalin, carbamazepine, phenytoin, and lamotrigine), and opioid analgesics have been widely recommended in the treatment of PS-TP [11,30,32] (see Table 1).

### 2.1. Tricyclic Antidepressants (TCAs)

One double-blinded, placebo-controlled trial with oral amitriptyline and carbamazepine for PS-TP showed that only amitriptyline had a significant reduction of pain when compared to placebo [5]. Amitriptyline and imipramine reduced thermal and mechanical hypersensitivity and pain in another study [13,33,34,35]. 

A few systematic reviews showed that amitriptyline and lamotrigine were the most effective in the treatment of PS-TP. Amitriptyline, lamotrigine, and gabapentin provide a more favorable efficacy and safety profile than other drugs, such as carbamazepine and phenytoin, but with low or very low evidence [29,31,35].

#### Role of Prophylactic Treatment

In the placebo-controlled study with 39 patients with PS-TP, a prophylactic/early starting treatment of PS-TP, which was diagnosed within the first day following stroke, had no significant beneficial effects when treated with amitriptyline [36].

### 2.2. Selective Serotonin Reuptake Inhibitors (SSRIs)

In an open-label, observational study with PS-TP, fluvoxamine had a significant pain relief in visual analogue scale (VAS) after the treatment of 4 weeks. Interestingly, the significant pain reduction was only in the patients who were treated within 1 year from stroke [37].

Another SSRI, fluoxetine, blocked mechanical hypersensitivity and pain but not thermal-related pain [33].

### 2.3. Serotonin–Norepinephrine Reuptake Inhibitors (SNRIs)

Duloxetine, a serotonin–norepinephrine reuptake inhibitor, is known to be effective against centralized neuropathic pain in multiple sclerosis [38].

In an open-label study with 37 patients, duloxetine also improved PS-TP, with a significant reduction of pain intensity. Twenty-six (70.3%) patients showed at least 30% reduction in numeric rating scale at the third week. Adverse events related with study withdraw were nausea, agitation, and somnolence [39].

### 2.4. Anticonvulsants

The second-line treatment for PS-TP includes anticonvulsants, e.g., gabapentin, pregabalin, carbamazepine, phenytoin, and lamotrigine [30,31,34,35].

Pregabalin is the most tested agent for the management of centralized neuropathic pain, but the evidence for its effectiveness is mixed. The safety, efficacy, and tolerability of pregabalin were also approved in a cohort study including 219 patients with PS-TP [32]. A few other cohort studies have also reported that pregabalin could be effective in the management of PS-TP as a monotherapy [40], as well as in combination with others [41]. An open-label randomized controlled trial with pregabalin and lamotrigine in 30 patients with PS-TP [42] suggested that both drugs were equally effective in the reduction of symptoms. However, the effectiveness of pregabalin in pain reduction has not yet been approved in a placebo-controlled RCT, therefore there is no high evidence. Although pain reductions at follow-up did not differ significantly between pregabalin and placebo, the improvements in secondary outcomes such as sleep and anxiety by pregabalin was helpful in the management of PS-TP [42].

Among the traditional medications, lamotrigine is the most effective anticonvulsant in the treatment of PS-TP. In several reviews, lamotrigine had the strongest evidence for the management of PS-TP, derived only from small randomized controlled trial [10,31,34,35,42].

One randomized double-blinded placebo-controlled study with two 8-week treatment periods separated by 2 weeks investigated the effectiveness of orally administered lamotrigine at 200 mg daily in 27 patients with PS-TP [43]. Global Pain Scores in the lamotrigine group were significantly reduced compared to the placebo. Although lamotrigine is safe, well-tolerated, and effective in different studies [30,31,40,41,43], a recent small-sized study suggested to discontinue the treatment with lamotrigine due to skin rashes [42].

In a prospective observational study, gabapentin, another anticonvulsant, at 300 mg twice daily, was effective in the management of PS-TP in 84 patients with thalamic stroke [44]. Additionally, the use of gabapentin has been reported individually in a few patients with PS-TP [31].

Neither levetiracetam nor carbamazepine has been found to be effective in the management of PS-TP [10,33,45]. A few clinical studies showed that carbamazepine had no sufficient effect in the treatment of PS-TP [5,10,33].

### 2.5. Other Medications

The third-line or other medications of PS-TP have not been well-assessed, but they could be individually used.

Opioid or opioid antagonist, medical cannabinoids, mexiletine, clonidine, and beta-blockers can be recommended as the third line treatment in individual cases with PS-TP. Intravenous infusions of lidocaine, often together propofol or ketamine, steroid, naloxone, as well as intrathecal baclofen or ketamine also showed their effectiveness in a few reports with acute severe pain of PS-TP [34,35,46,47,48].

In one recent systematic review with eight placebo-controlled studies, opioids, such as morphine, levorphanol, as well as competitive opioid-receptor antagonists including naloxone, showed only a slight analgesic effect in the treatment of PS-TP, with low evidence [46,49]. With low evidence, intravenously administered naloxone had no effect in pain reduction when compared with the placebo in the small-sized placebo-controlled study including 20 participants with PS-TP [49].

Medical cannabinoids [48] were effective in the management of chronic pain syndromes, e.g., multiple sclerosis; however, they have been not studied on poststroke thalamic pain (PS-TP) yet. In a 60-year-old woman with PS-TP, the treatment with medical cannabinoid showed pain relief [50].

Intravenous lidocaine together with propofol or ketamine are a well-known combinative infusion therapy for acute pain and also showed temporary pain relief in some patients with PS-TP, but because of the potential side effects, their long-term use could not be recommended [31].

Mexiletine, a natrium channel blocker, and an adjunctive to antidepressants, can be used individually, as with lidocaine; however, it has often been ineffective in clinical practice [25,35,47].

Modafinil, FDA-approved eugeroic can be used for the fatigue and medication overdosed headaches lasting over three months in patients with PS-TP. Modafinil is effective in both reducing fatigue while improving quality of life in them [51]. 

The local injection of botulinum neurotoxin A, well-known as a chemical neurolyser, which blocks the release of acetylcholine at the nerve terminal throughout the inhibition of the soluble N-ethylmaleimide-sensitive factor attachment receptor (SNARE) proteins, has also been shown as an effective therapy for therapy-resistant chronic peripheral (e.g., different neuralgia or peripheral neuropathic pain) as well as central neuropathic pain (e.g., poststroke pain, multiple sclerosis), and its mechanism of central effects in brain, including central afferent transport and transsynaptic movement of BoNT-A, has been discussed [52,53,54,55]. Therefore, this can also be further considered in the management of PS-TP.

## 3. Nonpharmacological Interventions of PS-TP

In the cases with intractable PS-TP, nonpharmacological interventions can be recommended as an important option, and appear to be effective in the treatment of PS-TP, but the evidence is relatively low, especially due to the limited numbers of participants. Additionally, invasive electrical brain stimulation can be followed by serious adverse events, which were recovered in most patients [56] (see Table 1).

In one recent systematic review [56] including 11 studies with 166 patients, the safety and effectiveness of different nonpharmacological interventions were assessed in the management of PS-TP. The pain symptom of PS-TP was significantly improved by precentral gyrus stimulation (*p* < 0.05), caloric vestibular stimulation (*p* < 0.01), transcranial direct current stimulation (*p* < 0.05), and bee venom acupuncture point injection (*p* < 0.01). Acupuncture (*p* > 0.05) and electroacupuncture therapies (*p* > 0.05) were not significantly effective for thalamic pain. Invasive motor cortex stimulation (not deep brain stimulation) was also effective for treating therapy-resistant PS-TP and appeared to be more effective than thalamic stimulation for reducing bulbar pain due to Wallenberg syndrome [56]. Deep brain stimulation (DBS) showed mixed effects in the improvement of pain as well as depression and anxiety in some patients with PS-TP. Some serious adverse events due to invasive brain stimulations, such as cerebrospinal fluid leak, hematoma, connective tissue growth above dura, vertigo, nystagmus, infection, and seizure, were reported in some cases, but most of them recovered [56]. 

### 3.1. Noninvasive Brain Stimulation

#### 3.1.1. Repetitive Transcranial Magnetic Stimulation (rTMS)

Repetitive transcranial magnetic stimulation (rTMS) as a noninvasive brain stimulation therapy has been often recommended in the treatment of PS-TP [30,57,58]. The rTMS is a safe and well-tolerated intervention [58]. The application of rTMS on the motor cortex for the treatment of PS-TP has been studied, mostly in small studies with low quality [58,59]. 

In an interesting study using diffusion tensor imaging fiber tracking, it was shown that the corticospinal tract as well as the thalamocortical tract plays a role in pain reduction by rTMS [60]. Moreover, the antalgic effect of rTMS in patients with PS-TP was more prominent in patients without depression [61]. Five daily sessions of rTMS over the motor cortex area in patients with refractory PS-TP resulted in pain relief lasting 2 weeks [62]. 

Another cohort study with the navigation-guided 5 Hz rTMS over the primary motor cortex corresponding to the painful hand suggested that the restoration of pathological cortical hyperexcitability might be one of the mechanisms of pain relief by deep rTMS in intractable PS-TP [63]. Using brain navigation, another small-sized (14 cases) cohort study found that the five sessions of rTMS with 2000 stimuli/10 Hz each session improved pain and thermal hypersensitivity, and this effect lasted for 4 weeks after the treatment, and the reduction of detection threshold for warm/cold sensation and pain relief were strongly correlated. Therefore, they suggested that pain relief by rTMS could be also from the reduction of hypersensitivity for noxious and thermal stimuli in the insula and the somatosensory and anterior cingulate cortices [64]. 

However, a double-blinded, randomized, sham-controlled trial including 23 patients with therapy-resistant PS-TP evidenced no significant analgesic effect of rTMS on the premotor cortex/dorsolateral prefrontal cortex [65].

Later, three recent sham-controlled studies promised its effectiveness again. The cohort study with maintaining rTMS once a week as an adjunct therapy suggested that it could help to relieve PS-TP [66]. Another study with rTMS on intractable lower limb pain of PS-TP showed the effectiveness with short-term pain relief [67]. One recent randomized sham-controlled study [57] including 40 patients with PS-TP assessed the effectiveness of 3-week rTMS treatment (10 Hz, 2000 stimuli) (n = 20) compared to a sham intervention (n = 20). Significant improvement was found for pain intensity in the rTMS group compared with the sham group already at 7 days (*p* < 0.001), and this effect lasted until the third week (*p* < 0.01). The serum level of brain-derived neurotrophic factor (BDNF), a neuronal modulator, was significantly higher in the treated group (*p* < 0.05). The applied rTMS over the area of the motor cortex for the upper extremity can effectively improve acute PS-TP, possibly by influencing cortical excitability and serum BDNF secretion [57]. 

Repetitive transcranial magnetic stimulation is a safe, well-tolerated, noninvasive therapy and can be strongly recommended in the treatment of intractable or therapy-resistant PS-TP, which has no sufficient effects of oral medications.

#### 3.1.2. Transcranial Direct Current Stimulation (tDCS)

An emerging treatment for intractable PS-TP is transcranial direct current stimulation (tDCS), a noninvasive brain stimulation, together with repetitive transcranial magnetic stimulation (rTMS) in clinical practice [59]. The safety and effectiveness of tDCS has been reported in many clinical studies for different neuropathic pain syndromes [68].

However, there has been only one sham-controlled, small-sized study [69] of tDCS treatment including 14 participants with PS-TP. The tDCS intervention (a 2-mA current intensity for 20 min per session) was performed with three sessions per week for a period of 3 weeks, and the anodal electrode was placed over primary motor cortex. The tDCS group (n = 7) showed a significant decrease of pain in the visual analogue scale (VAS) (*p* < 0.05) as well as thermal hypersensitivity (*p* < 0.05), while the sham group (n = 7) showed no statistically significant changes in time (*p* > 0.05). They suggested that tDCS improved sensory identification and thermal hypersensitivity, with a sufficient analgesic effect in stroke patients with PS-TP [69]. The evidence from this study is low, but nevertheless safe, well-tolerated, noninvasive interventions such as tDCS and rTMS in clinical practice can be recommended for the management of PS-TP, and future studies are required to approve the effectiveness as well as to standardize detailed treatment parameters. 

#### 3.1.3. Vestibular Caloric Stimulation (VCS)

In some patients with PS-TP, the pain decreased after VCS, which may rebalance the imbalance in the bilateral integration of thermal sensation, so called as the thermosensory disinhibition hypothesis, and this effect was due to the temporary activation of the parieto-insular vestibular cortex of VCS [22,70]. Vestibular caloric irrigation of the ear can lead to the activation of several areas in the contralateral hemisphere, including the insular cortex. The posterior insula is responsible for painful stimuli. Because of the phylogenetically and anatomical proximity of both the pain and vestibular area, VCS may improve the symptoms of PS-TP, which represents a pathological amplification of the thalamic posterior insular response to pain [24]. 

The VCS therapy, e.g., with cold/warm water, showed a temporary analgesic effect in some cases with PS-TP. One case series report with cold water vestibular caloric stimulation had favorable results in the treatment of PS-TP [11].

A single-blinded sham-controlled study including nine patients with PS-TP reported a significant immediate treatment effect of the cold-water VCS with pain reduction, and this result also supported the thermosensory disinhibition hypothesis after thalamic stroke [22].

One recent case report with a 57-year-old woman suffering from PS-TP showed the beneficial effect of VCS in the reduction of pain in the clinical as well as brain imaging data [70].

VCS is also a safe therapy for PS-TP and has shown a temporary effectiveness in pain reduction, despite limited evidence. Repeated VCS can be also recommended, if the irritation is tolerated, for the patients for severe intractable PS-TP.

### 3.2. Invasive Therapeutic Interventions of Brain Stimulation

#### 3.2.1. Epidural Motor Cortex Stimulation (EMCS)

Compared with other invasive brain stimulations, e.g., deep brain stimulation (DBS), epidural motor cortex stimulation (EMCS) is more frequently used because this intervention is safer and easier, and it has more indications, including PS-TP [71,72]. Based on the progress of neurosurgical techniques, it is a safe surgical intervention to implant a EMCS neurostimulator for the long-term electrical brain stimulation over the primary motor cortex [71,72] Additionally, the positive response of 10 Hz - rTMS in the preoperative test can predict the analgetic effect of EMCS [73].

The EMCS therapy for PS-TP has also been studied, but in small-sized cohort studies, and has shown sufficient pain relief lasting for up to 2 years [71,72,74,75].

EMCS was mostly effective for PS-TP [74], and sufficient pain reduction was obtained more frequently than in other stimulation therapies over the same site, e.g., 48% by EMCS compared to 7% by spinal cord stimulation and 25% by deep brain stimulation [76]. The recent case reports with EMCS therapy showed sufficient analgetic effect in five (71%) of seven patients with PS-TP, and this was also the highest efficacy in pain relief by EMCS in several neuropathic pains, such as atypical facial pain, postbrachial plexus avulsion injury pain, phantom pain, and pain in the syringomyelia [77]. One case report including six patients with PS-TP compared the analgesic effects of EMCS and deep brain stimulation, and both interventions showed their effectiveness in pain relief; however, there were no significant differences on the effect sizes between both interventions [78]. EMCS is safe and effective, but the infection after the surgical intervention was reported in several cases. Except that, no other serious adverse event has been reported [71,72].

#### 3.2.2. Deep Brain Stimulation (DBS) in PS-TP

DBS has been well-known as a safe and effective therapy in the management of many neurological diseases, especially essential tremors, other tremors, obsessive-compulsive disorder, neuropathic pain, traumatic brain injury, Tourette’s syndrome, and drug-resistant epilepsy [79]. 

Deep brain stimulation has also shown to be effective in over 50% of patients with PS-TP, and the stimulation to the thalamus alone showed less effect than other area stimulations in combination, e.g., periventricular/periaqueductal gray matter (PVG/PAG) or periventricular/periaqueductal gray matter (PVG/PAG) plus thalamus/internal capsule [78,80,81]. However, DBS, an invasive brain stimulation, was less effective or ineffective to PS-TP than other diseases or neuropathic pain syndromes in one cohort study [82].

These studies with mixed results were, however, with small-sizes or several cases, and therefore the evidence is poor [78,80,81,82] and further studies are necessary for better evidence of its safety and effectiveness on PS-TP. 

DBS is more invasive than other interventions in the management of PS-TP, and some serious adverse events, including the leak of cerebrospinal fluid, hematoma, postoperative infection, connective tissue growth above dura, vertigo, nystagmus, infection, and seizure, were reported in some cases [56]. Therefore, DBS therapy in the treatment of PS-TP can be recommended lastly, if other therapies are ineffective, and must be carefully applied after multidisciplinary consultation.

#### 3.2.3. Spinal Cord Stimulation

Spinal cord stimulation (SCS) was rarely recommended in PS-TP; however, in one cohort study, SCS improved with over 50% of pain reduction in 30% of patients with PS-TP [83]. 

The SCS therapy for PS-TP has been studied in a few cohort studies, which suggested that SCS can improve the long-term pain relief effect in some participants, but it was often ineffective in most patients with PS-TP [83,84,85]. All included studies were small-sized and noncontrolled with fair or poor quality, therefore further data for this intervention is required.

#### 3.2.4. Other Invasive Therapies

The direct electrical stimulation to the trigeminal ganglion and rootlets by using implanted electrodes had up to 50 percent pain relief of poststroke facial pain in a small-sized study of seven patients [86], and this intervention can be also considered for the PS-TP, including severe facial pain. 

Pituitary radiosurgery as a primary minimally invasive therapy option for patients with therapy-resistant PS-TP showed a high rate of initial efficacy in pain relief; however, in most of the pain reoccurred in 6–12 months. Its side effects were reported in 42% of patients, including anterior pituitary abnormalities, required hormonal replacement therapy, transient diabetes insipidus, transient hyponatremia, and clinical deterioration due to the worsening numbness, despite the reduction of pain [87].

Neurosurgery such as thalamotomy and mesencephalic tractotomy is a last option, if indicated, and a few reports have shown improved allodynia by this surgical intervention [25].

### 3.3. Other Nonpharmacological Therapies

Several studies assessed the effectiveness of acupuncture or electroacupuncture in PS-TP [11,71,88]. 

One therapy–comparative study with an acupuncture group (n = 32) and a pregabalin (75 mg, twice a day) group (n = 32) showed that acupuncture could effectively relieve the symptoms of PS-TP with long-term effect, and it is better than pregabalin [88].

Bee venom injection at acupuncture points reduced significantly the pain symptoms of PS-TP in visual analogue scores after three weeks compared with baseline and the control group [89].

However, the recent systematic review concluded that the included studies were with poor or fair quality and the outcomes were mixed with low evidence [30]. Perhaps acupuncture therapy can be recommended as an adjunctive treatment for PS-TP.

Cognitive-behavioral therapy was useful in the prevention of depression in patients with PS-TP [90]. Psychologic relaxation therapy should also be a part of adjunctive treatment [13]. 

## 4. Conclusions

Several poststroke pathologic changes in the processing and interpreting of sensory information, such as central imbalance, central disinhibition, central sensitization, other thalamic changes, and inflammatory responses on the affected neural pathway, may cause PS-TP, which is mostly therapy-resistant, constant, or timely unchanged neuropathic pain. The pharmacological treatments, including antidepressants (e.g., amitriptyline) and anticonvulsants (e.g., lamotrigine, pregabalin), which were suggested as the most favorable drugs from a few systematic reviews, are firstly recommended for PS-TP, and the safe and well-tolerated noninvasive interventions for local brain stimulation, such as rTMS, tDCS, and VCS, can be strongly recommended directly after the failed first-line medications in the treatment of intractable PS-TP. Other medications, including opioids (e.g., morphine) or opioid antagonist (e.g., naloxone), as well as medical cannabinoids, beta-blockers, and natrium-channel blockers (e.g., mexiletine), can be individually recommended depending on the effectiveness as well as the tolerance to the medications. Intravenous administration of lidocaine, often together with propofol or ketamine for the acute pain of PS-TP, and steroid infusion treatment in the early phase following the stroke in the cases with PS-TP, can be individually introduced. Intrathecal administration of baclofen or ketamine can be considered in the cases with other poststroke symptoms including spasticity-related pain.

EMCS as a safe and minimally invasive intervention that is strongly recommended in the management of intractable PS-TP. Other invasive interventions such as DBS, SCS, or other surgical treatments can be recommended lastly, if there is no better option, because they only have low evidence of their effectiveness and high risk of serious adverse events. 

## Figures and Tables

**Figure 1 diagnostics-12-01439-f001:**
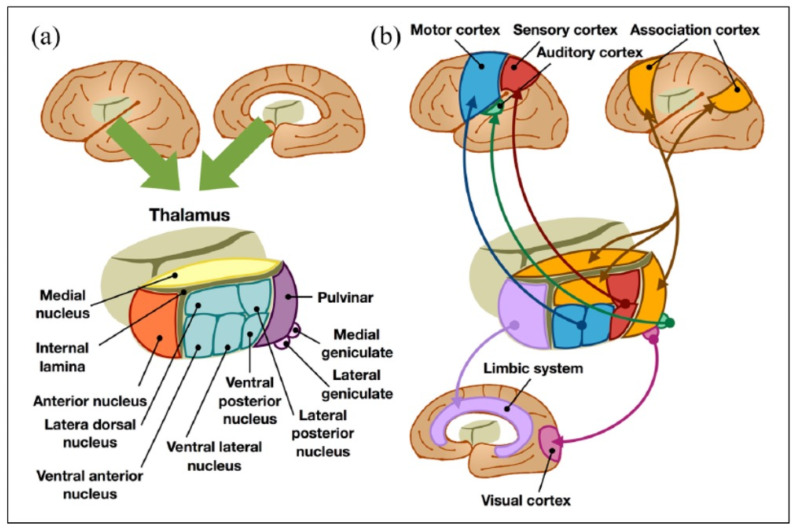
(**a**) Anatomical and (**b**) functional classification of the thalamus [reproduced from [15] the under the open access. Copyright © 2022 by SAGE Publications].

**Table 1 diagnostics-12-01439-t001:** The recommended pharmacological and nonpharmacological treatments for PS-TP.

Treatments	First/Second Recommendations	Third-Line and Others
Pharmacological	Antidepressants (amitriptyline *, imipramine, fluvoxamine, duloxetine, venlafaxine, trazodone), anticonvulsant (lamotrigine *, pregabalin *, gabapentin,) phenytoin	Oral carbamazepine, levetiracetam, opioid, medical cannabinoids, mexiletine, clonidine, modafinil, and beta-blocker; intravenous (opioid antagonist (naloxone), lidocaine, steroid, propofol), and intrathecal (baclofen, ketamine) administration, BoNT-A injection
Nonpharmacological	rTMS *, tDCS *, VCS *, EMCS	DBS, SCS, other surgical interventions, acupuncture, behavioral-, psychologic therapy

* Firstly recommended; rTMS: repetitive transcranial magnetic stimulation; tDCS: transcranial direct current stimulation; VCS: vestibular caloric stimulation; EMCS: epidural motor cortex stimulation; BoNT-A: botulinum neurotoxin A.

## Data Availability

Not applicable.

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
