# Peer review of "The Management of Poststroke Thalamic Pain: Update in Clinical Practice"

_diagnostics, 2022, doi:10.3390/diagnostics12061439_

Round 1

Reviewer 1 Report

this is a very interesting paper presenting the management of post-stroke thalamic pain.

line 30 "1. post-stroke thalamic pain" should be "1. Post-stroke thalamic pain"

line 127 "2. pharmacologic treatments of PS-TP" should be  "2. Pharmacologic treatments of PS-TP"

line 228 "non-pharmacological interventions of PS-TP" should be "Non-pharmacological interventions of PS-TP"

 line 300 "vestibular caloric stimulation (VCS)" should be "Vestibular caloric stimulation (VCS)"

The chapter "2.5. Other medications" should be more detailed and have a more structured form.

The chapter "4. Conclusions" should be more detailed and should propose detailed treatment management. 

Author Response

Dear Sir,

Thank you so much for your precious comments.

We revised and modified our manuscript after your comments.

line 30 "1. post-stroke thalamic pain" should be "1. Post-stroke thalamic pain"

I corrected it.

line 127 "2. pharmacologic treatments of PS-TP" should be  "2. Pharmacologic treatments of PS-TP"

I corrected it.

line 228 "non-pharmacological interventions of PS-TP" should be "Non-pharmacological interventions of PS-TP"

I corrected it.

 line 300 "vestibular caloric stimulation (VCS)" should be "Vestibular caloric stimulation (VCS)"

I corrected it.

The chapter "2.5. Other medications" should be more detailed and have a more structured form.

Your comment is correct. This section was modified. Please see the reversion.

The chapter "4. Conclusions" should be more detailed and should propose detailed treatment management. 

The conclusion was modified after your comment. Please see the reversion.

Thank you so much.

Best Regards

Reviewer 2 Report

This is a potentially interesting manuscript on central pain consequent to thalamic networks damage due to stroke. The only previous review articles on the same topic date 2017 (PMID: 28494691) and 2016 (PMID: 27175563). For this reason, I believe that a further manuscript is welcome. However, I found the following issues:

  • English requires correction throughout the manuscript.
  • In the abstract, line 13, indicate what “them” means (survivors?).
  • Line 30, Post.
  • Why is the reported prevalence (?) so variable (2.7-25%)? This should be commented on.
  • Another interesting point is the latency (months to years...). This suggests a long term plasticity or failure of an endogenous analgesic mechanism.
  • Line 37, check “was” (most commonly follows).
  • Line 38, check “any other”.
  • Line 40, check “had”.
  • Line 42, check “showed”.
  • The references reported in line 51 are not referring to a general description of somatosensory systems, other articles could be more appropriate (PMID: 1810518).
  • Figure 1 is misplaced; it should be placed where first cited.
  • Isn’t there a role for the reticular thalamic nucleus? This relay is important in regulating the input/output oscillatory signals from thalamus (PMID: 10938331).
  • Check for the inconsistent use of abbreviations (line 90).
  • Some description of mechanisms involved in drugs used for central pain is missing (for instance, PMID: 20298965, PMID: 32751761).
  • Table 1 should also be placed in a different line (146).

Author Response

Dear Sir,

Thank you so much for your precious comments.

We revised and modified our manuscript after your comments.

  • English requires correction throughout the manuscript.

The manuscript was corrected and modified by a native speaker.

  • In the abstract, line 13, indicate what “them” means (survivors?).

Yes, you are right. “them” means survivors.

  • Line 30, Post.

It was corrected.

  • Why is the reported prevalence (?) so variable (2.7-25%)? This should be commented on.

You are right.

The prevalence of post-stroke thalamic pain was highly variable from many clinical studies, mainly due to their different duration of study follow-up observation as well as the different distribution of the stroke etiology and lesions, detailed therapeutic approaches.

We added the following sentence after your comment:

The prevalence of PS-TP was highly variable, because of different study duration as well as clinical characteristics of stroke population included in clinical studies e.g., stroke etiology, distribution of stroke lesions, clinical management.

  • Another interesting point is the latency (months to years…). This suggests a long-term plasticity or failure of an endogenous analgesic mechanism.

You are right. We discussed on the part of its pathogenesis.

  • Line 37, check “was” (most commonly follows).

It was corrected.

  • Line 38, check “any other”.

We deleted it.

  • Line 40, check “had”.

It was corrected.

  • Line 42, check “showed”.

I checked it.

  • The references reported in line 51 are not referring to a general description of somatosensory systems, other articles could be more appropriate (PMID: 1810518).

You are right absolutely. PMID: 1810518 is added after your comment. However, we tried to focus on the pathogenesis of the post-stroke thalamic pain.

  • Figure 1 is misplaced; it should be placed where first cited.

Figure 1 is replaced.

  • Isn’t there a role for the reticular thalamic nucleus? This relay is important in regulating the input/output oscillatory signals from thalamus (PMID: 10938331).

Yes, you are right. We described about it at the last part of the pathogenesis of PS-TP as followed:

On the other side, NLRP3 inflammasome at thalamus lesion strengthens inflammatory response of microglia at the same time. Persisting inflammatory process can induce GABAergic changes in thalamus reticular neurons to inhibit the functions of ventral basal interneurons. These can result on the PS-TP. [17] Several factors e.g., stress and skin temperature aggravate PS-TP. [11]

  • Check for the inconsistent use of abbreviations (line 90).

We checked them.

  • Some description of mechanisms involved in drugs used for central pain is missing (for instance, PMID: 20298965, PMID: 32751761).

After your comment we modified the section, especially on several recent medications against central neuropathic pain e.g., botulinum toxin A. However, most of the fist- and second line medications have already been well-known for many years.

  • Table 1 should also be placed in a different line (146).

It was replaced.

Thank you so much for your precious time and great comments.

Best Regards

Round 2

Reviewer 2 Report

All my comments have been addressed.